# Comparative analysis of pre-Covid19 child immunization rates across 30 European countries and identification of underlying positive societal and system influences

**Marco Cellini**[1], **Fabrizio Pecoraro**[1]*, **Michael Rigby**[2], **Daniela Luzi**[1]

**1** National Research Council, Institute for Research on Population and Social Policies (CNR-IRPPS), Rome, Italy, **2** School of Social, Political and Global Studies and School of Primary, Community and Social Care, Keele University, United Kingdom

* f.pecoraro@irpps.cnr.it

## Abstract

This study provides a macro-level societal and health system focused analysis of child vaccination rates in 30 European countries, exploring the effect of context on coverage. The importance of demography and health system attributes on health care delivery are recognized in other fields, but generally overlooked in vaccination. The analysis is based on correlating systematic data built up by the Models of Child Health Appraised (MOCHA) Project with data from international sources, so as to exploit a one-off opportunity to set the analysis within an overall integrated study of primary care services for children, and the learning opportunities of the 'natural European laboratory'. The descriptive analysis shows an overall persistent variation of coverage across vaccines with no specific vaccination having a low rate in all the EU and EEA countries. However, contrasting with this, variation between total uptake per vaccine across Europe suggests that the challenge of low rates is related to country contexts of either policy, delivery, or public perceptions. Econometric analysis aiming to explore whether some population, policy and/or health system characteristics may influence vaccination uptake provides important results—GDP per capita and the level of the population's higher education engagement are positively linked with higher vaccination coverage, whereas mandatory vaccination policy is related to lower uptake rates. The health system characteristics that have a significant positive effect are a cohesive management structure; a high nurse/doctor ratio; and use of practical care delivery reinforcements such as the home-based record and the presence of child components of e-health strategies.

## Introduction

Even before the Covid-19 pandemic, declining rates of child immunization in Europe make it imperative to analyse the complex mix of factors influencing uptake, as well as the goals set by the European Vaccine Action Plan (EVAP) 2015–2020 whose aims have not been met [1].

**Data Availability Statement:** The data underlying the results presented in the study are available from World Health Organization (WHO), World

Bank and Eurostat. As cited in the manuscript authors will provide a datasheet on zenodo that integrates all data adopted for the econometric analysis. The dataset has been published on Zenodo and is available here at the folloing DOI: 10.5281/zenodo.6619113. The dataset is cited in the paper as reference [40].

**Funding:** The author(s) received no specific funding for this work.

**Competing interests:** The authors have declared that no competing interests exist.

Thus, further actions are essential to achieve equitable access and improve surveillance and monitoring based on high-quality data [2, 3].

Research and policy recommendations by international organizations tend to focus on barriers related to complacency and confidence [4–6], or on structural and organizational components of national health systems [2, 7–9]. What is too often overlooked is that vaccination should be considered and delivered as a part of integrated child health services [10–12] and viewed from a parental and child perspective as to desirability, accessibility, and barriers [13, 14]. Too often well-intentioned policy initiatives are aspirational, rather than built on solid data [15], or ignore the specific issues of child aspects [16–19]. Variation in coverage between countries and across vaccines calls into question the extent to which national contexts of policy interventions, public perceptions, socio-economic features, and specificity in the wider health system and vaccination delivery, are direct influences needing specific consideration.

This study provides a first macro-level analysis within the European Union (EU) and European Economic Area (EEA) into child immunization embedded in a societal and system context, analysing variations across vaccines and across countries. As part of this, comparative time series for key vaccine coverage have been compiled for 30 countries over 10 years. From this baseline, an econometrics analysis was then performed to look at possible contextual variables that may influence vaccination coverage. The study takes advantage of systematic data built up by the Models of Child Health Appraised (MOCHA) Project (available at: www.childhealthservicemodels.eu) and correlates them with international data sources, exploiting a one-off opportunity to set analysis within an overall integrated study of primary care services for children, and the learning opportunities of the 'natural European laboratory'. Thus, population vaccination uptake data are analysed uniquely in a composite of time series, policy, and structural contexts.

## Materials and methods

### Immunization data

The year 2018, as the culmination year of the MOCHA project, was taken as the anchor year for all data sources in order to seek maximum contemporaneousness. Data on immunization coverage have been gathered from the World Health Organization (WHO) website [20] that collects country-reported administrative data annually through the WHO/UNICEF joint reporting process. Based on the availability of data on immunization coverage in the then 30 EU/EEA countries, the following vaccines were included in the analysis:

- first and third dose of Diphtheria, Tetanus, Pertussis-containing vaccine (DTP1 and DTP3).

- third dose of Hepatitis B containing-vaccine (HEPB3).

- third dose of Haemophilus influenzae type b-containing vaccine (HIB3).

- third dose of inactivated polio-containing vaccine (POL3).

- third dose of pneumococcal conjugate-containing vaccine (PCV3).

- first and second dose of measles-containing vaccine (MCV1 and MCV2).

For all of the above the measure provides the number of infants who have received the vaccination related to the population of surviving infants. The cohort is composed of children between 12 and 24 months of life, except for MCV2 where the cohort composition depends on the national schedule [21].

## Data on population, health system, and policies

We then considered several characteristics of countries' populations, their health systems, and their policies which could be hypothesised to have a determinant effect upon immunization including parental motivation.

In order to analyse such characteristics, the following variables were considered. Table 1 provides the summary of their typology, unit of measure and information on data collection (year range considered, data source and date of access to the relevant resources).

### Population characteristics

- *GDP per capita*: Represents the gross domestic product per capita. Data from the World Bank [22] database.

- *Gini index*: Measures countries' level of inequality. Data from the SWIID (Standardized World Income Inequality Database) produced by Solt [23].

- *Tertiary education engagement*: Represents the gross enrolment ratio, namely the ratio of total enrolment regardless of age, to the population of the age group that normally corresponds to the level of education shown. Thus, it is not specifically focussed on measuring parental education, but the tertiary educational uptake of the (mainly younger adult) population. Data from the World Bank [24] database.

- *Child proportion*: The share of children and young people aged 0–19 out of the total population. Data from Eurostat [25]–given the absence of child population data [26], we aggregated the number of persons for age classes 0–5, 5–9, 10–14, and 15–19 then divided by the total population.

- *Rural population*: Measures the share of citizens living in rural areas. Data retrieved from the World Bank [27] database.

### Health system characteristics

- *Nurses/doctors ratio*: Represents the ratio between the total number of nurses and the total number of doctors serving the whole population in all healthcare settings. It is calculated on

**Table 1. Detailed information on data typology, unit of measure, years of coverage, source and date of access.**

| Variable | Type of variable | Unit of measure | Year(s) of coverage | Source and access link | Date of access |
|---|---|---|---|---|---|
| GDP per capita | Continuous | US dollars | 1991–2017 | World Bank | 2021 Aug 12 |
| Gini Index | Continuous | 0–100 index | 1991–2017 | Solt, 2016 | 2021 Jul 12 |
| Tertiary education enrolment | Continuous | Percentage of target popn. | 1991–2017 | World Bank | 2021 Aug 16 |
| Child proportion | Continuous | Percentage of total popn. | 1991–2017 | Eurostat | 2021 Aug 16 |
| Rural population | Continuous | Percentage of total popn. | 1991–2017 | World Bank | 2021 Aug 16 |
| Nurse/doctor ratio | Continuous | Ratio with range 0 to n | 1991–2017 | WHO - Doctors WHO - Nurses | 2021 Aug 16 |
| Decentralization | Dichotomous | 0–1 dummy | 2017 | European Union Committee of the Regions | 2021 Aug 16 |
| Type of primary care expertise | Dichotomous | 0–1 dummy | 2016 | Blair, Rigby and Alexander, (2017) | 2021 Aug 16 |
| Mandatory vaccination | Dichotomous | 0–1 dummy | 2010 | Bozzola et al., 2010 | 2021 Aug 16 |
| Child health strategy | Dichotomous | 0–1 dummy | 2016 | Blair et al., 2019 | 2021 Aug 16 |
| Child e-health strategy | Dichotomous | 0–1 dummy | 2016 | Kühne and Rigby, 2016 | 2021 Aug 16 |
| Home-based record | Dichotomous | 0–1 dummy | 2018 | Rigby et al., 2020 | 2021 Aug 16 |

the basis of number of nurses [28] and number of doctors [29] available in the WHO data-base. We calculated and adopted this ratio since both the number of doctors and the number of nurses showed a high variability across countries.

- *Decentralization*: A categorical variable identifying the degree of centralization/decentraliza-tion of national health systems. It is coded based on a European Union analysis [30], and it takes five distinct values: centralized; mostly centralized; operatively centralized; partially decentralized; and decentralized defined as:

  0. Centralized—all the power, responsibility and functions are with the central government or are deconcentrated, i.e., are given to entities at the territorial level which represent the central level.

  1. Mostly centralized—most of the power, responsibility and functions are with the central government, but lower levels of elected government still have a minor role in relation to health expenditure.

  2. Operatively decentralized—the central government has an important role within the health management system, but some operative functions are held by lower levels of the elected government.

  3. Partially decentralized—some of the power, responsibility and functions for health are transferred/devolved from the central government to lower, elected levels of government. The central government still has a role within the health management system, the impor-tance of this role varying depending on the level of devolution.

  4. Decentralized—except for some main framing conditions, the power, responsibility, and functions for health are not with the central government but with lower, elected levels of government.
  To estimate the effect of the different categories, in the regression we added them as sepa-rate dummy variables [31].

- *Type of Primary Care expertise*: Identifies whether the type of doctor who provides primary care for children is a community paediatrician or a general practitioner. The variable is coded as a dummy derived from data compiled by the MOCHA study [32], with a value of 1 if within a country there is a community paediatrician service (solely or alongside general practitioners) and value of 0 if there is no community paediatrician availability.

## Policy characteristics

- *Mandatory vaccination*: A dummy variable identifying those countries in which one or more vaccines are mandatory by law, taking a value of 0 if none of the vaccines are mandatory, and 1 if at least one vaccine is mandatory. It does not assess the rigour, if any, with which a country enforces this policy. The classification has been based on Bozzola et al. [33].

- *Child health strategy*: Identifies the presence of strategies for children and adolescents within the national health systems of the countries. The variable is coded based on information from the MOCHA project [34], with value 0 if a country does not have such a strategy, and 1 if it has a strategy. It does not assess the content, resourcing, or impact of the strategy [11].

- *Child e-health strategy*: Identifies the presence of specific aspects considering children and adolescents within national e-health strategies, coded based on Kühne and Rigby [35], with a value of 1 if a country considers children in its e-health strategy and of 0 otherwise.

- *Home-based record (HBR)*: This variable identifies the presence of home-based records within countries. It is coded based on Rigby, Deshpande and Namazova-Baranova [8] with a value of 1 if a country utilises home-based records comprehensively including immunization and of 0 otherwise. Within individual countries HBRs are also known as Personal Child Health Record, Parent Held Record, or as MutterKindPass (i.e., Mother-Child Passport) in German speaking countries [36].

## Analytic approaches

Countries' immunization coverage was analysed calculating the average values of vaccines within countries, while the relevant variability was analysed computing the coefficient of variation [37]. We took 95% as the target uptake threshold for all vaccines analysed.

Time series for DTP1–DTP3 and MMR1–MMR3 were comparatively analysed in terms of vaccination and country differences from 2009 to 2018.

An econometric analysis assessed the effect of these independent variables on vaccination coverage.

To do so, we employed a panel regression model to match the longitudinal nature of the data. The baseline model takes the form of Eq 1.

*Eq 1*: *Baseline panel regression model*

$$Y_{it} = \beta_0 + \beta_1 X_{it} + \beta_2 X_{it} + \ldots + \beta_k X_{it} + \mu \tag{1}$$

where *y* represents the dependent variable, $\beta_0$ the constant term, *X* represents the independent variables and $\mu$ represents the error term.

Substituting the terms with the variables employed in the analysis, the equation became:

*Eq 2*: *Estimates equation model*

*Average coverage$_{it}$*

$$
\begin{aligned}
&= \beta_0 + \beta_1 GDP\ per\ capita_{it} + \beta_2 GINI\ index_{it} + \beta_3 Tertiary\ education_{it} \\
&+ \beta_4 Child\ proportion_{it} + \beta_5 Rural\ population_{it} + \beta_6 Nurses/doctors\ ratio_{it} \\
&+ \beta_7 Decentralized_{it} + \beta_8 Paediatrician\ lead_{it} + \beta_9 Mandatory\ vaccination_{it} \\
&+ \beta_{10} Child\ health\ strategy_{it} + \beta_{11} Child\ e-health\ strategy_{it} \\
&+ \beta_{12} Home-based\ records_{it} + \beta_{13} Country\ dummies + \mu
\end{aligned} \tag{2}
$$

Due to the presence of variables such as the level of decentralization that within countries take the same value along all the years considered, we were not able to add fixed effects to our regression. Nevertheless, to control for the presence of country-specific factors affecting vaccine coverages, we added a country dummy variable.

## Results

### Vaccination coverage and its variability

Table 2 reports the vaccination coverage rates across 30 countries for the eight vaccines considered in this study. It shows the variability and average in vaccination rates between countries, and across vaccines within the same country. Variability was analysed computing the coefficient of variation (CoV) as the ratio of the standard deviation to the average. The CoV is widely adopted to express the precision and repeatability of an assay as it facilitates comparison between data sets with different units or widely different means [37]. Data are presented as a heatmap according to whether the rate per vaccine per country is higher than 95% (green

**Table 2. 2018 vaccination coverage by vaccine and by country.**

| Vaccine: | DTP1 | DTP3 | HEPB3 | HIB3 | POL3 | PCV3 | MCV1 | MCV2 | Average | CoV |
|---|---|---|---|---|---|---|---|---|---|---|
| **Country** | | | | | | | | | | |
| Austria | 90 | 85 | 85 | 85 | 85 | | 94 | 84 | 86,9 | 4,3% |
| Belgium | 99 | 98 | 97 | 97 | 98 | 94 | 96 | 85 | 95,5 | 4,7% |
| Bulgaria | 94 | 92 | 85 | 92 | 92 | 88 | 93 | 87 | 90,4 | 3,6% |
| Croatia | 98 | 93 | 93 | 94 | 94 | | 93 | 95 | 94,3 | 1,9% |
| Cyprus | 99 | 99 | 97 | 97 | 97 | 81 | 90 | 88 | 93,5 | 7,0% |
| Czechia | 98 | 96 | 94 | 94 | 94 | | 96 | 84 | 93,7 | 4,8% |
| Denmark | 97 | 97 | | 97 | 97 | 96 | 95 | 90 | 95,6 | 2,7% |
| Estonia | 93 | 92 | 93 | 92 | 92 | | 87 | 88 | 91,0 | 2,7% |
| Finland | 99 | 91 | | 91 | 91 | 88 | 96 | 93 | 92,7 | 4,0% |
| France | 99 | 96 | 90 | 95 | 96 | 92 | 90 | 80 | 92,3 | 6,4% |
| Germany | 98 | 93 | 87 | 92 | 93 | 84 | 97 | 93 | 92,1 | 5,1% |
| Greece | 99 | 99 | 96 | 99 | 99 | 96 | 97 | 83 | 96,0 | 5,7% |
| Hungary | 99 | 99 | | 99 | 99 | 99 | 99 | 99 | 99,0 | 0,0% |
| Iceland | 97 | 91 | | 91 | 91 | 90 | 93 | 95 | 92,6 | 2,8% |
| Ireland | 98 | 94 | 94 | 94 | 94 | 90 | 92 | | 93,7 | 2,6% |
| Italy | 98 | 95 | 95 | 94 | 95 | 92 | 93 | 89 | 93,9 | 2,8% |
| Latvia | 97 | 96 | 96 | 96 | 96 | 82 | 98 | 94 | 94,4 | 5,4% |
| Lithuania | 95 | 92 | 93 | 92 | 92 | 82 | 92 | 92 | 91,3 | 4,3% |
| Luxembourg | 99 | 99 | 96 | 99 | 99 | 96 | 99 | 90 | 97,1 | 3,3% |
| Malta | 99 | 97 | 98 | 97 | 97 | | 96 | 95 | 97,0 | 1,3% |
| Netherlands | 97 | 93 | 92 | 93 | 93 | 93 | 93 | 89 | 92,9 | 2,3% |
| Norway | 99 | 96 | | 96 | 96 | 94 | 96 | 93 | 95,7 | 2,0% |
| Poland | 98 | 95 | 91 | 95 | 87 | 60 | 93 | 92 | 88,9 | 13,6% |
| Portugal | 99 | 99 | 98 | 99 | 99 | 98 | 99 | 96 | 98,4 | 1,1% |
| Romania | 94 | 86 | 93 | 86 | 86 | | 90 | 81 | 88,0 | 5,2% |
| Slovakia | 99 | 96 | 96 | 96 | 96 | 96 | 96 | 97 | 96,5 | 1,1% |
| Slovenia | 97 | 93 | | 93 | 93 | 60 | 93 | 94 | 89,0 | 14,5% |
| Spain | 97 | 93 | 94 | 94 | 93 | 93 | 97 | 94 | 94,4 | 1,8% |
| Sweden | 99 | 97 | 92 | 97 | 97 | 97 | 97 | 95 | 96,4 | 2,1% |
| United Kingdom | 98 | 94 | | 94 | 94 | 92 | 92 | 88 | 93,1 | 3,2% |
| Average | 97,4 | 94,5 | 93,3 | 94,3 | 94,2 | 88,9 | 94,4 | 90,4 | | |
| CoV | 2,2% | 3,7% | 4,0% | 3,6% | 3,9% | 11,5% | 3,2% | 5,5% | | |

Red cells coverage < 90%, yellow cells coverage between 90% and 95%, green cells coverage > 95%. White cells no data available.

cells), lower than 90% (red cells) or between 90 and 95% (yellow cells). The analysis of vaccination rate coverage was done considering two dimensions–country and vaccine–to enable inter-country and inter-vaccine analyses.

In 2018, only four countries (Hungary, Malta, Portugal, and Slovakia) reported a coverage higher than 95% for all vaccines, Hungary having the highest (99%). However, Sweden and Luxembourg have only one vaccine with coverage lower than 95%, while in contrast four countries (Austria, Bulgaria, Estonia, Romania) have a coverage lower than 95% for all vaccines, of which Austria is the country with the lowest average coverage (86,9%).

Viewing coverage rates by vaccine across Europe, there is no vaccine with a coverage of 95% or more in all countries. The vaccine with a coverage of 95% or more in the greatest number of countries is DTP1 (26 countries), while in contrast PCV3 and MCV2 reach 95%

coverage in only 7 countries. The vaccine with the most consistently high coverage is DTP1 with no country reporting below 90%, while a coverage rate for MCV2 below 90% is reported in 12 countries. Other than for DTP1 with its uniformly high uptake, and PCV3 which is only given in 24 countries, the uptake average is quite similar between countries, with an average uptake range per vaccine across the 30 countries of between 12 and 19 percentage points (and between 12 and 14 percentage points per vaccine excluding >also MCV2).

Figs 1 and 2 show the relationship between the coverage rate and the relevant variability of country and vaccine dimensions. In both analyses a high average coverage is associated with a low level of variability. In particular, there is a small set of high-performing countries with a high level of immunization coverage for almost all vaccines. The matching result in the inter-vaccine comparative analysis shows vaccines with high average of coverage with high values for most countries, though some vaccines with low average immunization coverage neverthe-less have some countries exceeding the 95% target.

While there is no vaccine with a low coverage rate in all the analysed countries, with the exception of PCV3 there is less variation between total uptake across Europe per vaccine (range 90,4–97,4, CoV 2,2–5,5) than between countries (range including MCV2 86,9–99,0, CoV 0,0–14,5), suggesting that low rates are related to country contexts of either policy, deliv-ery, or public demand and acceptance.

## Time series analysis—DTP vs. MCV vaccines

Trends in vaccination coverage are analysed from 2009 to 2018 considering the first and third doses of DTP and the first and second doses of MCV, respectively, taken as examples of the overall best and the worst performing vaccines. This can detect change over time as well as issues concerning the subsequent doses identified in some studies [38, 39] as a crucial point

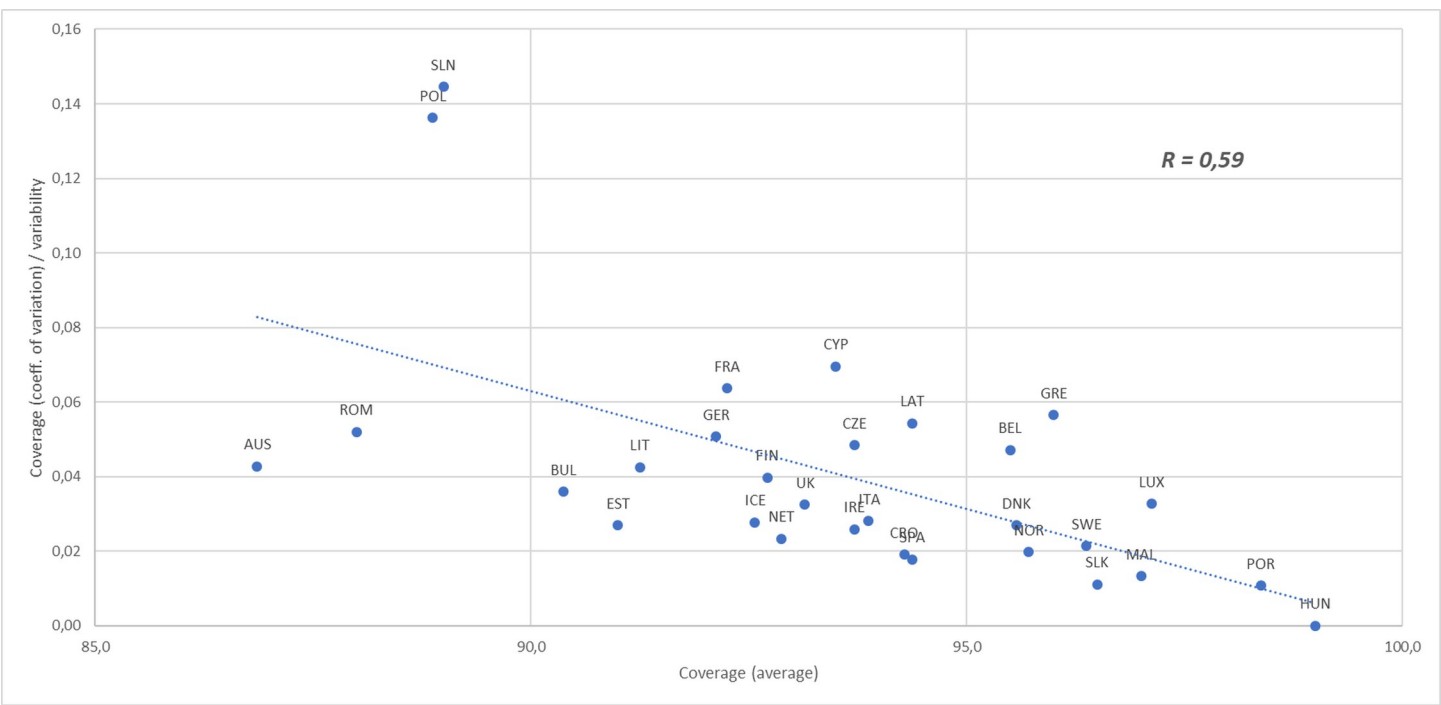

**Fig 1. Scatterplot diagram reporting the correlation between the average vaccination coverage and the vaccination coverage variability computed at country level (i.e. inter-country analysis).**

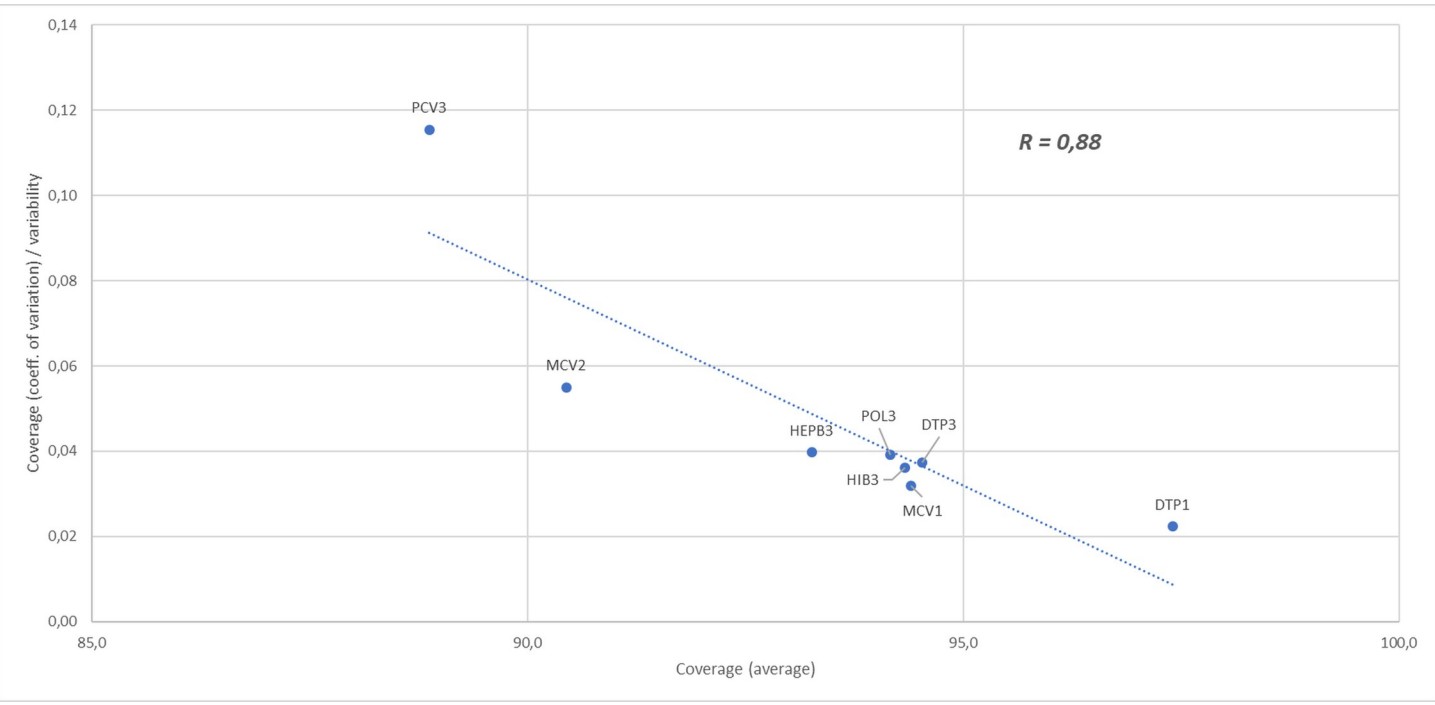

**Fig 2. Scatterplot diagram reporting the correlation between the average vaccination coverage and the vaccination coverage variability computed at vaccine level (i.e. inter-vaccine analysis).**

for the vaccination uptake. Moreover, the identification of similar patterns across countries could guide further analyses to detect whether other common contextual factors influence vaccination coverage.

As shown in Figs 3 and 4, some similar patterns can be detected, confirming the variability also along the time series considered. For the two doses of each vaccine, three countries (Greece, Hungary, Luxemburg) have no differences in DTP1 and DTP3, and one (Hungary) in MCV1 and MCV2. This equal performance has a positive effect for these countries which constantly reach the target of 95% vaccination coverage.

The lower coverage of the subsequent dose in both DTP and MCV, present in most countries, also follows different trends over time. In some cases, the two trajectories have almost constant values outlining a synchronized pattern. This is the case in both DTP and MCV in three countries (Netherlands, Germany and United Kingdom), while in Portugal this is true for DTP and in Belgium for MCV. Intriguingly, these groups of countries have very different health systems, and are largely not contiguous. When the difference between the two doses is lower, the decrease of the second dose does not prevent reaching the vaccination coverage target, as in Portugal. In Belgium, where the target has been constantly reached in DTP1, DTP3 and MCV1, the lower rate of MCV2 (83%) may signal a specific issue in catching up with children at an older age.

Variations in these patterns highlight years in which the subsequent doses start decreasing, but only for one of the analysed vaccines—for Finland, Croatia, Poland, Slovakia and Slovenia for DTP3, and in the Czech Republic for MCV2, making it difficult externally to interpret which factors may have adversely influenced a generally stable vaccination coverage such as DTP. The opposite trend, more evident in the increase of MCV2, may indicate that specific efforts toward its uptake have been successful, as in France where MCV2 has progressively

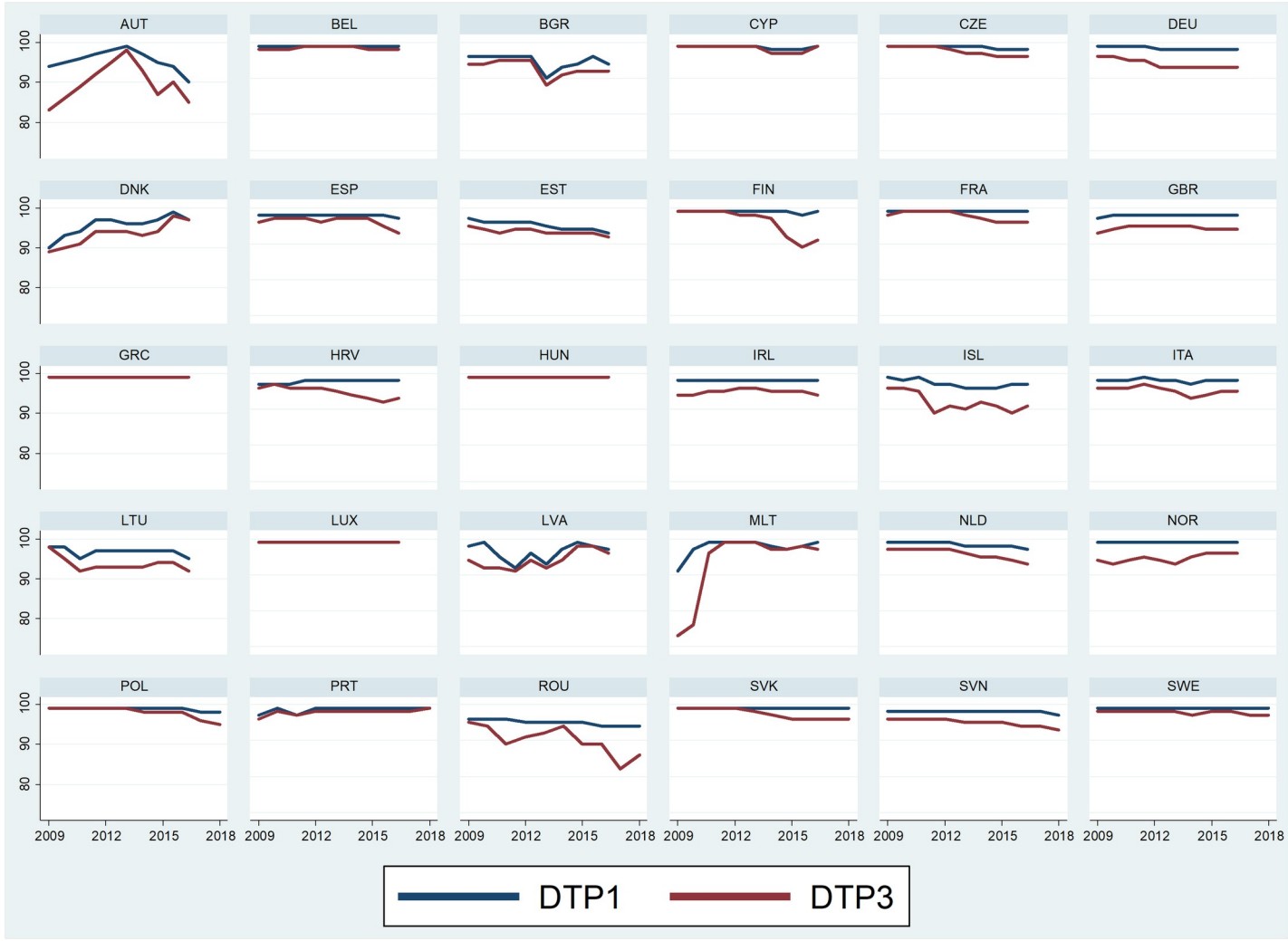

**Fig 3. Vaccination coverage (%) for the first and third doses of the DTP vaccine for each country in the period 2009–2018.**

improved since 2010 by some 20 percentage points. In Italy where both DTP1 and DTP3 have constant coverage rates, the increase of the two MCV doses started in 2015 may coincide with reactions to the measles outbreak and the changes in mandatory vaccination policy in 2017.

However, despite the differences between doses and vaccines, the majority of countries showed relatively continuous time trends, though specific peculiar trends can be detected in a few countries. For instance, Austria has an increasing parallel trend for all vaccines until 2014 and then has a significant decrease, in particular for DTP. While the rate in 2009 was lower than 95% for all vaccines and doses, in 2014 DTP and MCV1 were all within the target, but the subsequent decrease from 2015 has led Austria back out of the target for all vaccines. Another peculiar example is Norway, with discordant trends between the two doses of MCV: while MCV1 increased over time, MCV2 decreased. This led the first dose to be on target in 2018 and the second dose to be outside the target.

If patterns of trends are difficult to identify and compare, the time series analysis highlights a worrying decrease in five countries–Bulgaria, Estonia, Lithuania, the Netherlands, and Poland—in particular for MCV in which the coverage target is not met anymore, and to a

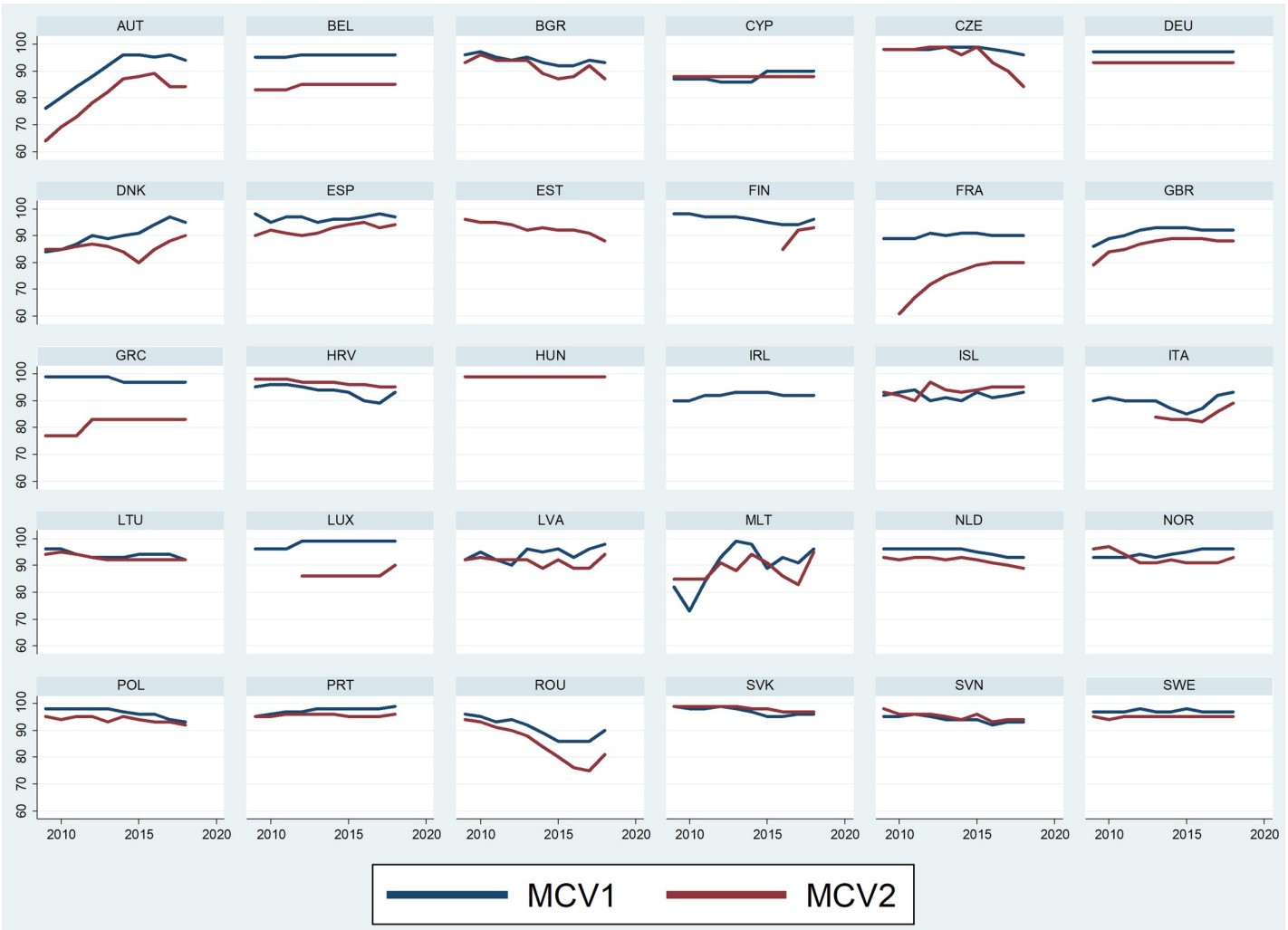

**Fig 4. Vaccination coverage (%) for the first and second dose of MMR vaccine for each country in the period 2009–2018.** Note that Ireland provides data only for MCV1, while no information is available for MCV2.

lesser extent also in DTP though not compromising the target except in Estonia. Indeed, while Estonia met the target in 2009, all vaccine coverage was below 95% in 2018. Conversely, positive steps in uptake are noticeable in Denmark, where since 2008 the low coverage of both DTP and MCV has improved (+5% in MCV). Besides France and Italy, Greece, Malta, Spain and UK show progressive increases in uptake, especially for MCV2.

## Econometric analysis

Having examined the country time series for vaccination coverage, the study moved to consideration of the immunisation context by analysing the population, health system and policy aspects already described. Table 3 reports summary statistics of the variables. We present the full table of country-specific characteristics data used for the analysis on line as a Zenodo file [40]. The sample is composed of 25 countries–Denmark, Finland, Iceland, Norway, and the United Kingdom having to be excluded because of lack of relevant data (the complete list of the countries and years covered is reported in Table 1 in the S1 File). This constitutes a

**Table 3. Summary statistics of the variables adopted in the econometric analysis reporting the number of observations, the average and standard deviation as well as the minimum and maximum values.**

| Variable: | Country/year Observations | Average | Standard Deviation | Min | Max |
|---|---|---|---|---|---|
| Average coverage | 368 | 93.5 | 5.6 | 67.0 | 99.0 |
| GDP per capita | 368 | 24612.9 | 17628.8 | 3582.89 | 110162.1 |
| GINI Index | 368 | 31.6 | 3.7 | 22.0 | 39.600 |
| Tertiary education engagement | 368 | 59.5 | 18.1 | 9.2 | 136.603 |
| Children proportion | 368 | 22.5 | 3.0 | 18.1 | 33.135 |
| Rural population | 368 | 30.5 | 11.0 | 2.0 | 49.246 |
| Nurses/doctors ratio | 368 | 2.02 | 0.76 | 0.61 | 5.824 |
| Decentralization | 368 | 2.03 | 1.23 | 0 | 4 |
| Type of Primary Care expertise | 368 | 0.277 | 0.448 | 0 | 1 |
| Mandatory | 368 | 0.473 | 0.500 | 0 | 1 |
| Child health strategy | 368 | 0.541 | 0.499 | 0 | 1 |
| Child e-health strategy | 368 | 0.541 | 0.499 | 0 | 1 |
| Home based record | 368 | 0.802 | 0.399 | 0 | 1 |

technically unbalanced panel of 25 countries for the timespan 1991–2017 with a total number of 368 country/year observations, as a result of the missing values within countries and years.

Table 4 reports the results of the reported regression. For each variable standard errors, relevant levels of significance as well as the beta coefficients are reported.

**Table 4. Results of the regression analysis reporting, for each model and each independent variable, the standard errors and the level of significance in parentheses as well as the relevant beta coefficients.** All models were run considering the average vaccination coverage as dependent variable.

| Variable: | Beta coefficient | Standard error | Significance level |
|---|---|---|---|
| GDP per capita (per 1000 USD) | 0.26 | 0.112 | ** |
| Gini index (per 100) | -2.4 | -12.9 | |
| Tertiary education engagement | 0.12 | 0.03 | *** |
| Children proportion | 0.16 | 0.19 | |
| Rural population | -0.07 | 0.19 | |
| Nurses/doctors ratio | 2.53 | 0.78 | *** |
| 1.Mostly centralized | 12.8 | 7.7 | * |
| 2.Operatively decentralized | 33.6 | 14.9 | ** |
| 3.Partially decentralized | -8.6 | 4.3 | ** |
| 4.Decentralized | 15.8 | 13.7 | |
| Provision of primary care community paediatrician | -8.9 | 9.9 | |
| Mandatory vaccination | -7.4 | 2.9 | ** |
| Child health strategy | -28.8 | 7.2 | *** |
| Child e-health strategy | 19.1 | 7.2 | *** |
| Home based record | 9.4 | 3.8 | ** |
| Constant | 67.2 | 9.4 | *** |
| Number of country/year observations | 368 | | |
| Number of countries | 25 | | |
| Country dummies | YES | | |
| $R^2$ | 0.55 | | |

*** p<0.01,

** p<0.05,

* p<0.1

The regression results show that with respect to population characteristics GDP per capita and tertiary education engagement show a positive and significant effect, while Gini index and rural population show no significant effect. Concerning health system characteristics, results show that a higher nurse doctor ratio is significantly associated with higher vaccination rates. At the same time, the analysis highlights that having a mostly centralized or an operatively decentralized health system has a positive and significant effect on vaccination coverage, while having a partially decentralized system shows a significant negative effect, suggesting that clarity and simplicity of operational structure are the optimum enabling factors rather than structure itself. Note that the decentralization level was added as a series of dummy variables, but the first category (i.e., centralised system) was automatically dropped from the regression to avoid multicollinearity issues. In addition, the regression indicates that the presence of mandatory vaccination and of national child health strategies have a negative and significant effect on vaccination coverage, while the presence of a national child e-health strategy and the employment of home-based records have a positive and significant effect.

## Limitation of the study

The first limitation concerns the size of the sample for the econometric analysis, in which the absence of key observations for some of the countries and/or years resulted in the construction of an unbalanced panel that may affect the estimations. This lack of data forced us to reduce the sample from the target 30 countries to 25 countries, namely excluding Denmark, Finland, Iceland, Norway, and the United Kingdom, due to data gaps (see Table 1 in the S1 File).

Moreover, variables that identify the presence of policy features (Child health strategy, Child e-health strategy and Home-based record) were included as binary indicators of the attention focussed on child issues; analysis of the strength of such strategies and their implementation would require qualitative investigations not available, but the simple associations are strong.

## Discussion

The descriptive analysis of vaccination coverage in the decade to 2018, based on the average values and their coefficients of variation, made it possible to capture differences across the 30 EU/EEA countries and across the eight most common vaccines. The inter-country analysis showed that, besides a small set of high-performing countries, the level of variability can be differently associated with countries showing vaccination coverage rates that meet the target of 95% only with some vaccines, or with countries showing low uptake rates related to almost all vaccines. In the inter-vaccine analysis, besides DTP1 with the highest number of countries reaching the 95% vaccination target, low rates are limited to PCV3 and MCV2. The analysis of country time series of DTP and MCV vaccines confirms high variability, making it difficult to outline similar patterns between doses and across countries. Contrasting with this, the lower range of coefficient of variation across vaccines suggests that the challenge of low rates is related to country contexts.

However, the ability to juxtapose these immunisation data with economic and demographic data, and with more specific data from the child health policy research of the MOCHA project, gives much greater richness of analysis and creates some stable and important pointers to potentially valuable further analytic topics. The macro level econometric and structural factor analysis showed how specific aspects appeared to influence vaccination coverage. Though perforce based on snapshot national data, some results need to be taken into consideration. Concerning the populations' characteristics, it is noteworthy that GDP per capita and the level of educational engagement are positively and significantly associated with higher vaccination

coverage, while the inequality index and the children proportion show no significant effects. Concerning the share of the population living in rural areas, with a negative but non-significant effect, most of the international literature identifies that a higher share of rural population within countries is associated with worst performance in terms of vaccination coverage. The contrasting lack of significance in our results might be explained by European countries' health systems being usually well developed, where the coverage is guaranteed widely in rural and in urban areas, whereas much of the literature on this topic covers a wider range of countries and levels of development.

Regarding health system characteristics, the analysis shows the nurses/doctors ratio to have a positive and significant effect—the higher the number of nurses compared to the number of doctors, the higher the vaccination coverage, even though these data relate to the whole health system and not just to prevention or services for children. This may indicate that a positive nurse/doctor ratio leads to healthcare overall being more holistic, reflecting person-centred values and focussing away from purely biomedical clinical interventions and illness focus. However, due to lack of detailed information this remains a hypothesis that needs to be tested in future studies. Unfortunately, the source data do not show to what extent nurses are employed in preventive care or in vaccination activities specifically. Therefore, we cannot know whether the effect of the nurse/doctor ratio to vaccination coverage is due to nurses' active role in vaccination activities, or to other aspects such as promotion of a family-centred or integrated child life-course approach. This lack of information highlights the need to design and implement a better collection of data about nurses' participation and their role in preventive activities, without which it is not possible to enumerate and understand the mechanisms through which they positively influence overall infant vaccination coverage.

Another characteristic of national health system considered in the analysis is the level of centralization/decentralization of the system itself. It is interesting to note that while having either a mostly centralized or an operatively decentralized system is significantly associated with higher vaccination rates, having a partially centralized system show a significantly negative effect. This may indicate that those systems presenting mixed decentralization characteristics are worst suited to carry out vaccination activities effectively, possibly due to conflicts or confusion in the allocation of duties, communication, and accountability not being well defined or efficient.

Moreover, and importantly as it is at first sight counter-intuitive, the analysis shows that the presence of mandatory vaccination policies is matched to a negative and significant relationship to vaccination coverage. The influence of mandatory vaccination on vaccination uptake is a controversial issue in the literature, with some authors arguing that it helps to increase uptake [41], while other authors claiming the contrary [42, 43]. The negative and significant effect shown in our analysis may confirm the claims of the latter; or alternatively, may suggest that the introduction of mandatory vaccination policies has been done as a policy panic measure when constructive measures and delivery structures fail. Mandatory vaccination policies, in fact, vary widely among the countries considered and in the rigour of implementation [44], so that while in some countries the consequences of not vaccinating are quite high, such as unvaccinated children not being permitted to go to school, in other countries the penalties are far less serious, with only some kind of economic fine. Additional considerations are the possible alienation experienced by those parents that are more hesitant toward vaccines, and those having anti-vaccination attitudes, further polarizing their negative or sceptic view on vaccines.

Concerning the other policy characteristics, the analysis shows that while having a child health strategy has a negative and significant effect on vaccination coverage, having a child e-health strategy and employing home based records have positive and significant effects. The different findings of association between child health strategy and child e-health strategy

initially may seem perverse. However, it is very likely that the effect of those kind of strategies does not depend on the simple presence or absence but rather on the content of the strategies themselves and how they are implemented. In other words, the discrepancies between the two kinds of strategies could depend on the fact that they consider vaccination in different ways, if at all, and with different priorities. However, to be able to disentangle such effects further research is needed. It is likely that child health strategies will be broad, and will cover many important issues such as child mental health services, support for those with chronic conditions and needing long-term care, and may be addressing acknowledged service deficits, with the result that apparently simpler issues such as immunization will be considered (possibly wrongly) as not being in need of such in depth treatment. By contrast, e-health strategies may focus much more on transactional and recording issues and immunization will be an important area for such data management modernisation, as examples show [34]. Lastly, the positive effect shown by the deployment of home-based records containing vaccination data confirms the claims of the WHO [45] and their role in encouraging parental involvement and responsibility.

## Conclusions

This analysis shows that far more complex determinants drive vaccination rates in Europe than merely anti-vaccination sentiments. It underscores the importance of taking a societal and a user focused approach, as well as recognising the effect of GDP and tertiary education engagement levels. At national policy level, GDP and access to higher education are issues wider than the health sector, but have an effect on it, in line with the WHO Health in all Policies approach [46]. Zdunek et al. [47] have differentiated the proximal and distal influences on children's health, but our results seem to suggest that the whole system has an influence, in that parents as proximal agents are better empowered where there is higher GDP and better access to tertiary education, and that health professionals as agents are more influential in immunisation uptake where there is a higher nurse-orientated culture, hence the distal factors enable the proximal.

National policies seeking to modify parental behaviour by statute or regulation, which are potentially aggressive or punitive, by making vaccination mandatory, seem counter-effective. By contract, constructive policies and initiatives, the emphasis of WHO on holistic life-course approaches to children's preventive health services [11], and the work of Bedford et al. [13] and of the Expert Panel on Effective Ways of Investing in Health [14], seem to be salient in highlighting other factors including access and barriers [48]. At this overall health policy level, there seems broad success at addressing the challenges of inequality, and of rural service delivery, as these seem to have been counter-balanced and overcome in most EU and EEA countries. Also at system level, avoiding incomplete decentralisation, having a high nurse to doctor ratio, utilising home-based records for children, and innovation in digital health systems focussed on child health, seem to foster higher vaccination rates.

This study was the result of a one-off opportunity to bring together three very different data sources for child immunisation in 30 European countries. It can be read in terms of policy implications. In this sense, what the analysis seems to suggest is that a country willing to improve its vaccination uptake should implement policies aimed at increasing citizens' educational level, and to invest in e-health strategies incorporating elements of immunization services–partnership and 'public health in all policies' approaches. At the same time, according to our results, countries should rethink their policies on mandatory vaccination and, moreover, their overall health system should be either purely centralized or decentralized, as mixed forms seem to negatively influence vaccination uptake. Lastly, to reach a better understanding on the

relation between vaccination uptake and the health staff employed in the national health systems, countries should stimulate studies on the respective volume and different roles played by doctors and nurses in national health systems, and in immunization delivery in particular.

The baseline data for this study relate to the period just prior to the Covid-19 pandemic. That major public health challenge will have changed service structures and disturbed public attitudes, not least with regard to vaccination. However, the findings should still be helpful in the challenge of moving childhood immunisation forward, by showing the factors and structures which have been most positively influential in the immediate past.

## Supporting information

**S1 File.**
(DOCX)

## Author Contributions

**Conceptualization:** Marco Cellini, Fabrizio Pecoraro, Michael Rigby.

**Data curation:** Marco Cellini, Fabrizio Pecoraro, Michael Rigby.

**Formal analysis:** Marco Cellini, Fabrizio Pecoraro, Daniela Luzi.

**Funding acquisition:** Daniela Luzi.

**Investigation:** Marco Cellini, Fabrizio Pecoraro, Daniela Luzi.

**Methodology:** Marco Cellini, Fabrizio Pecoraro, Michael Rigby, Daniela Luzi.

**Supervision:** Michael Rigby, Daniela Luzi.

**Validation:** Fabrizio Pecoraro, Daniela Luzi.

**Visualization:** Fabrizio Pecoraro.

**Writing – original draft:** Marco Cellini, Fabrizio Pecoraro, Michael Rigby, Daniela Luzi.

**Writing – review & editing:** Marco Cellini, Fabrizio Pecoraro, Michael Rigby, Daniela Luzi.

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
