## [Editor Report · Decision Letter 0]

4 Feb 2022

PONE-D-21-30876Comparative analysis of child immunization rates across 30 European countries and identification of underlying positive influencesPLOS ONE

Dear Dr. Pecoraro,

Thank you for submitting your manuscript to PLOS ONE. After careful consideration, we feel that it has merit but does not fully meet PLOS ONE’s publication criteria as it currently stands. Therefore, we invite you to submit a revised version of the manuscript that addresses the points raised during the review process.

The study has merits but, in its current version, the manuscript has some major issues that have to be solved regardless of any review. Thus, before sending the manuscript out for the peer review process, I suggest the authors to address these issues. This will substantially fasten the review process.1.Details should be provided on when data on each country characteristics (such as policies) have been extracted. They may vary and we need to know when the assessment was made.2.The description of the Tables is poor overall. There are many crucial details that have not been explained and must be explained. E.g. in Table 1, what is CoV? "Antigen" should be replace by "Vaccine", in Table 2 what 368 observations stands for? No units of measures are given, 3 decimals are not needed, min./max. can be cut, in Table 3, there is no explanation of what is the dependent variable, the title average coverage is wrongly placed, in place of "covariates" or "variables", six decimals are surreal (change the unit of the variable in thoushands), there is no indication that the numbers are referred to beta coefficients, the building of the final model creates confusion (are we searching the best model to evaluated an association, or are we making a statistical exercise?). Why non significant variables have been retained in the final model has to be explained, and once the final, most complete model has been fit, there is no need at all to leave the results of the intermediate models, unless there are very major reasons. As it currently stands, although the aim is straightforward, it is confusing and very difficult to understand. Again, in the results there is a long description of the three models, which is most likely trivial. Please discuss the results of the best model.3.Overall, the writing is fair, but can be improved. The topic is complex, and the authors should try to make the sentences as short as possible. As it currently stands, it is sometimes too complex for the reader to understand the concepts. Some sentences are simply impossible to understand and must be rewritten. E.g. "The inter-country analysis showed that, besides a small set of high-performing countries, the level of variability can be differently associated with countries showing vaccination coverage rates that meet the target of 95% only with some vaccines, or with countries showing low uptake rates related to almost all vaccines".Other sentences are written in Italian and then translated, but must be improved. E.g. "gives much

greater richness of analysis and creates some stable and important findings" or "Though perforce based on snapshot national data". Overall, the writing of manuscript should be revised in depth, keeping in mind that the discussion is very long and only truly essential concepts should be maintained.4.Please address errors such as "Error! Reference source not found" 5.All the figures do not have clear explanations and, as they currently stand, they are impossible to understand.After these issues are addressed, the manuscrpt will proceed to the review process. The study is valid.

We look forward to receiving your revised manuscript.

Kind regards,

Lamberto Manzoli, M.D., M.P.H.

Academic Editor

PLOS ONE

https://journals.plos.org/plosone/s/file?id=ba62/PLOSOne_formatting_sample_title_authors_affiliations.pdf"
---

## [Author Response · Author response to Decision Letter 0]

24 Feb 2022

Dear Editor,

We thank you for your email regarding our submission to the PLOS one journal. We are grateful for your suggestions and the comments.

We have answered the comments raised in a point-by-point format addressing any concerns. Comments are reported in red in the response to the editor letter enclosed in this submission. The manuscript has considerably improved by these modifications. We have highlighted the changes in the main text using track changes.

We sincerely hope that these revisions will speed up the review process considering that we have submitted this paper on October 2021, more than five months ago. 

Best regards,

Fabrizio Pecoraro

---

## [Decision Letter · Decision Letter 1]

26 Apr 2022

PONE-D-21-30876R1Comparative analysis of pre-Covid19 child immunization rates across 30 European countries and identification of underlying positive societal and system influencesPLOS ONE

Dear Dr. Pecoraro,

Thank you for submitting your manuscript to PLOS ONE. After careful consideration, we feel that it has merit but does not fully meet PLOS ONE’s publication criteria as it currently stands. Therefore, we invite you to submit a revised version of the manuscript that addresses the points raised during the review process.

 Please acknowledge that I entirely disagree with the reviewer statements on the lack of time for obligation to show some effects on uptake. Three years are absolutely sufficient to see an effect, and they are indeed the amount of time that Governments typically set to perform an evaluation. Given this, please address the minor issues raised by the reviewer and reduce the emphasis on the conclusions, acknowledging that we cannot have individual data and any conclusion can only be preliminary.

We look forward to receiving your revised manuscript.

Kind regards,

Lamberto Manzoli, M.D., M.P.H.

Academic Editor

PLOS ONE

Journal Requirements:

Reviewers' comments:

Reviewer's Responses to Questions

**Comments to the Author**

1. If the authors have adequately addressed your comments raised in a previous round of review and you feel that this manuscript is now acceptable for publication, you may indicate that here to bypass the “Comments to the Author” section, enter your conflict of interest statement in the “Confidential to Editor” section, and submit your "Accept" recommendation.

Reviewer #1: (No Response)

2. Is the manuscript technically sound, and do the data support the conclusions?

Reviewer #1: No

3. Has the statistical analysis been performed appropriately and rigorously? 

Reviewer #1: I Don't Know

4. Have the authors made all data underlying the findings in their manuscript fully available?

Reviewer #1: Yes

5. Is the manuscript presented in an intelligible fashion and written in standard English?

Reviewer #1: Yes

6. Review Comments to the Author

Reviewer #1: It is well established that vaccine uptake is context and vaccine specific and high uptake depends on numerous inter related complex factors. Using available data the authors set out to explore which factors influence vaccine uptake across 30 European countries.

The findings of this analysis at three levels showed the positive impact of high GDP and tertiary education and young population along with a cohesive management structure, high doctor/nurse ratio, e-health strategies and home based records but the negative impact of mandatory vaccination.

I found this paper somewhat perplexing as there seems to be a number of leaps to conclusions that are not always borne out by the evidence.

For example, the authors found that a high nurse/doctor ratio had a positive effect and argue that this may result in healthcare being more holistic, reflecting person centred values and focussed away from purely biomedical interventions. They acknowledge that the data on nurse/doctor ratio does not allow the extent to which nurses are employed in preventive care or vaccination activities and recommend better information is needed in this respect. This is a fundamental question - could this conclusion lead to countries deciding to employ fewer doctors?!. However, an important issues is that in many countries, vaccination is doctor led rather than nurse led - this information should be easily available.

However, the issue over which I have most comment is the conclusion that mandatory vaccination has a negative relationship with vaccination coverage. Key to this finding is how long the mandatory vaccination policy has been in place. In some of the countries included in this analysis, mandatory vaccination was only introduced or requirements expanded in 2017/2018, the year taken in this project as the ‘anchor year’, in response to wide spread measles outbreaks, giving no time for the effects of mandation. The lower vaccine uptake in these countries may thus have resulted in the introduction of mandation and is not the cause of the low uptake.

Overall, I am not clear about the take home message from this paper other than richer countries, with cohesive management structures have higher vaccination rates. How would a country wishing to improve its vaccination uptake act on these findings? The study is an impressive use of data, but I do not feel takes us much further in understanding the determinants of vaccine uptake.

There are a number of minor details which require explanation for an international readership or correction:

Child e-health strategy, home based records – are both defined, but would benefits from further explanation. For example, in UK, home based records are known as Personal Child Health Records.

Page 16, line 331 – presume this should be ‘immunisation’ data not ‘immunological’

7. PLOS authors have the option to publish the peer review history of their article (what does this mean?). If published, this will include your full peer review and any attached files.

Reviewer #1: No

---

## [Author Response · Author response to Decision Letter 1]

9 Jun 2022

We would firstly like to thank the Editor and the Reviewer for their useful, pertinent, and insightful comments and suggestions. The issues raised by the Reviewer have been addressed throughout this new version of the paper. 

Following the detailed responses to the issues raised by the two Reviewer.

“For example, the authors found that a high nurse/doctor ratio had a positive effect and argue that this may result in healthcare being more holistic, reflecting person centred values and focussed away from purely biomedical interventions. They acknowledge that the data on nurse/doctor ratio does not allow the extent to which nurses are employed in preventive care or vaccination activities and recommend better information is needed in this respect. This is a fundamental question - could this conclusion lead to countries deciding to employ fewer doctors?!. However, an important issues is that in many countries, vaccination is doctor led rather than nurse led - this information should be easily available.”

The employment of the nurses/doctor ratio was based on the willingness to understand whether a higher number of nurses compared to the number of doctors was associated with higher vaccination rates within countries. Indeed, this was based on the implicit assumption that nurses concur in vaccination, and more broadly preventive care, activities. Unfortunately, we do not know whether this is the case or not, since there is not available information on the role performed by nurses within the different health systems considered in the analysis. For these reasons, in this new version we further clarified and made more explicit that the results concerning this variable are only preliminary and that there is the need to perform further research to understand the different roles in vaccination activities that nurses perform in the different countries (page 6 line 119; page 20 line 383). 

However, the issue over which I have most comment is the conclusion that mandatory vaccination has a negative relationship with vaccination coverage. Key to this finding is how long the mandatory vaccination policy has been in place. In some of the countries included in this analysis, mandatory vaccination was only introduced or requirements expanded in 2017/2018, the year taken in this project as the ‘anchor year’, in response to wide spread measles outbreaks, giving no time for the effects of mandation. The lower vaccine uptake in these countries may thus have resulted in the introduction of mandation and is not the cause of the low uptake.

We coded the “mandatory” variable based on the work of Bozzola et al. (2010). In particular, we coded it as a dummy variable taking the value of 0 (in all the years analysed) if no mandatory vaccination has ever been in place in a certain country and taking value of 1 otherwise. This choice was dictated by the fact that we employed the average value of several vaccine’s uptake rather than focusing on a single vaccine and that our interests was to test whether the presence of mandatory vaccinations within a country was associated with higher or lower rates. In other words, we did not assess the lag between the policy implementation and the changes in vaccination rates rather the direction of the relation between the presence of at least one mandatory vaccine, at aggregated level, and the average vaccination rates.

Overall, I am not clear about the take home message from this paper other than richer countries, with cohesive management structures have higher vaccination rates. How would a country wishing to improve its vaccination uptake act on these findings? The study is an impressive use of data, but I do not feel takes us much further in understanding the determinants of vaccine uptake.

Following your comment, in this new version of the paper we added a paragraph explicitly describing the implication of our results in terms of policy (page 22 line 452).

There are a number of minor details which require explanation for an international readership or correction:

Child e-health strategy, home based records – are both defined, but would benefits from further explanation. For example, in UK, home based records are known as Personal Child Health Records.

Page 16, line 331 – presume this should be ‘immunisation’ data not ‘immunological’

We clarified the meaning of Child health strategy and home-based records. We also did a proof reading of the entire document.

---

## [Editor Report · Decision Letter 2]

28 Jun 2022

Comparative analysis of pre-Covid19 child immunization rates across 30 European countries and identification of underlying positive societal and system influences

PONE-D-21-30876R2

Dear Dr. Pecoraro,

We’re pleased to inform you that your manuscript has been judged scientifically suitable for publication and will be formally accepted for publication once it meets all outstanding technical requirements.

Kind regards,

Lamberto Manzoli, M.D., M.P.H.

Academic Editor

PLOS ONE

Additional Editor Comments (optional):

All issues have been addressed.
---

## [Editor Report · Acceptance letter]

8 Jul 2022

PONE-D-21-30876R2 

Comparative analysis of pre-Covid19 child immunization rates across 30 European countries and identification of underlying positive societal and system influences 

Dear Dr. Pecoraro:

I'm pleased to inform you that your manuscript has been deemed suitable for publication in PLOS ONE. Congratulations! Your manuscript is now with our production department. 

Kind regards, 

on behalf of

Prof. Lamberto Manzoli 

Academic Editor

PLOS ONE